# HLA-class II genes association with multiple sclerosis: An immunogenetic prediction among multiple sclerosis Jordanian patients

**Sawsan I. Khdair**[1]*, **Lubna Al-Khareisha**[1,2], **Osama H. Abusara**[1], **Alaa M. Hammad**[1], **Alaa Khudair**[3]

**1** Faculty of Pharmacy, Al-Zaytoonah University of Jordan, Amman , Jordan, **2** Department of Pharmacy, Al-Bashir Hospital, Amman, Jordan, **3** Faculty of Dentistry, Jordan University of Science and Technology, Irbid, Jordan

* sawsan.khdair@zuj.edu.jo

## Abstract

Multiple sclerosis (MS) is an inflammatory autoimmune disease affecting the central nervous system (CNS). The pathogenesis of MS is characterized by neuronal axonal degeneration and demyelination. Among the genes that raises MS risk are the HLA-class II genes. The goals of this study were to investigate the role of the *HLA-DRB1* and *HLA-DQB1* genes (for the first time) in Jordanian MS patients and their association with MS disease. The association of these genes with other clinical features, such as optic neuritis, sensory impairment, and brainstem symptoms in MS patients was investigated as well using PCR-SSP techniques. Our findings indicated an association between *HLA-DRB1 *03:01* ($Pc$ = 0.01) and *HLA-DRB1 *04:01* ($Pc$ = 0.004) alleles with Jordanian MS patients. In addition, a significant linkage between *HLA-DRB1 *15:01* and *HLA-DQB1 *06:01* alleles ($Pc \leq 0.001$ and $Pc$ = 0.012, respectively) were presented among Jordanian MS patients with optic neuritis compared to Jordanian MS patients without optic neuritis. Moreover, *HLA-DQB1 *05:01* and *HLA-DQB1 *06:02* alleles ($Pc \leq 0.001$ and $Pc$ = 0.006, respectively) was found to be related with sensory impairment in MS patients. Additionally, *HLA-DRB1 *07:01* allele indicates a positive correlation in MS patients with brainstem symptoms ($Pc < 0.001$). Moreover, our results indicated that there is no association on the *HLA-DRB1* ~ *HLA-DQB1* haplotype level and MS disease. Knowing the genes that are linked to MS, they may facilitate MS diagnosis, prevention, and treatment at earlier stage. Also, these results may serve in the development of more potent therapeutic regimens for MS and its related complications, such as optic neuritis, sensory impairment, and brainstem symptoms.

## 1. Introduction

Multiple sclerosis (MS) is an inflammation in which the nerve cells in the central nervous system (CNS) are affected leading to the disease state. Demyelination and neuroaxonal damage are the hallmarks of MS, which cause a variety of signs and symptoms, such as optic neuritis and physical disability [1]. Young adults are more susceptible to MS, in which females are impacted almost three times as frequently as males [2]. Worldwide, the predictable number of people with MS is over 1.8 million [3].

**Data availability statement:** All relevant data are within the paper and its Supporting Information files.

**Funding:** The project was fully funded from Al-Zaytoonah University of Jordan grants (Grants numbers: 31/11/2020-2021 and 58/17/2022-2023). The funders had no role in study design, data collection and analysis, decision to publish, or preparation of the manuscript.

**Competing interests:** There are no competing interests to declare.

Environmental and genetic factors can both affect the likelihood of developing the disease. As for the environmental factors, the most well-known include vitamin D status, smoking, and Epstein-Barr virus [4]. Numerous studies have reported that there are genetic association between MS and major histocompatibility complex (MHC) genes, such as class II Human Leukocyte Antigen (HLA)-DR alleles being of great interest [5–7]. A study by the International Multiple Sclerosis Genetics Consortium has documented that there are more than 200 risk genetic variants that contribute to MS, but the HLA alleles remain the strongest being linked to MS [8]. In addition, a longitudinal review study which include 72 published articles (1993-2004) was completed by Human Genome Epidemiology has showed that MHC class II region, notably individuals with *HLA-DRB1 *15:01* allele, was in high risk to MS [9,10].

Antigen presentation on the surface of various cells can be due to MHC class II genes. These cells include dendritic cells, macrophages, and other immune cells. Collectively, these cells are known as antigen presenting cells (APCs). In autoimmune disease, MHC class II molecules -inappropriately- have the ability to be expressed on other cell types, such as astrocytes and oligodendrocytes in the brain [11–13]. Nevertheless, the mechanism of susceptibility to MS via MHC class II genes in unknown. However, one of the suggested mechanisms that explain the role of *HLA-DRB1 *15:01* allele in MS pathogenesis is that the Myelin Basic Protein (MBP), myelin oligodendrocyte glycoprotein, and proteolipid protein are three primary potential antigens for MS. Human MBP has received the most attention in studies that deal with the immune response against myelin antigens. Several studies (based on the structural theory) mentioned that the expression of *HLA-DRB1 *15:01* allele in MS patients had an immunodominant response to residues 84-102 of human MBP. Within the residues 85-99 of MPB; valine residue at location 89 and phenylalanine residue at location 92 were discovered to be located in the P1 and P4 anchor residues of the HLA-DRB1*15:01 peptide binding groove [10,14,15]. In a recent study, *HLA-DR *15* allele subtypes have been shown to work as an autoreactive T cell repertoire, in which the *HLA-DRB1 *15* allele's immunopeptidome presenting the foreign peptides and self-antigens are same as to autoreactive CD4 + T cells in MS [16].

MS affects many people in different geographical areas. For example, in North America, Western Europe, and Australasia, there are claims to be more than 100 cases per 100,000 individuals [17]. In terms of MS risk zones, countries of the Middle East and North Africa (MENA) region are considered part of the low- to moderate-intensity zone [18], and there is a scarcity of information of MS prevalence in Arab countries including Jordan.

Our study is the first one in terms of investigating the association of *HLA-DRB1* and *HLA-DQB1* allele polymorphisms and *HLA-DRB1 ~ HLA-DQB1* haplotypes with MS disease among Jordanian MS patients compared to healthy participants. In addition, we studied the linkage of *HLA-DRB1* and *HLA-DQB1* allele polymorphisms and *HLA-DRB1 ~ HLA-DQB1* haplotypes with different clinical features of MS, such as optic neuritis, sensory impairment, and brainstem symptoms in Jordanian MS patients.

## 2. Materials and methods

### 2.1. Sample collection

A total of 75 healthy participants representing the control group and 65 MS patients were involved in our study. The inclusion criteria for this study were that all volunteers should be Jordanians and not relatives. A questionnaire was distributed among healthy participants for the purpose of fulfilling their inclusion and exclusion criteria. A family history of autoimmune diseases was exclusion criteria for healthy participants. MS patients were all Jordanians and were enrolled from neuro-medicine clinic at Al-Bashir Hospital, Amman, Jordan from

February 2022 to November 2022. Whereas the exclusion criterion is that for patients who have other autoimmune diseases apart from MS. MS patients' information were obtained from their records at the neuro-medicine clinic at Al-Bashir Hospital, Amman, Jordan. MS patients were diagnosed according to McDonald diagnostic criteria [19]. About 3 ml of blood was collected in ethylenediaminetetraacetic acid tubes (EDTA) from each volunteer after signing a consent form. The Ministry of Health, Amman, Jordan granted its approval for the conduction of the study (Institutional Review Board number: 263/2021).

## 2.2. DNA extraction and HLA genotyping

The DNA was extracted from the 3 mL blood sample collected from individuals using the Wizard® Kit (Promega, USA) according to manufacturer's protocol. Briefly, it involves a four-stage technique to extract DNA from buffy coat of white blood cells after centrifugation. These are accomplished by the Cell Lysis Solution and the Nuclei Lysis Solution to lyse the red blood cells and white blood cells along with their nuclei, respectively, and then a salt-precipitation step to precipitate and remove the cellular proteins while the high molecular weight genomic DNA being present in the solution. Lastly, the genomic DNA is concentrated and desalted using an isopropanol solution. The purity of the DNA sample was assessed using a Nano-Drop spectrophotometer (Quawell, Canada). The amount of DNA used was around 45 ng and its purity was determined at the wavelengths A260/A280 nm, which were between 1.8 and 2.0. Genetic Constitution of *HLA-DRB1 and HLA-DQB1* alleles was achieved via PCR-SSP using One Lamda (generic and high resolution) Micro SSP™ Typing Trays (Thermo Fisher Scientific Inc., USA) according to manufacturer's protocol.

## 2.3. Statistical analyses

SPSS version 26 software (IBM Analytics, USA) was used to carry out the statistical analyses as discussed before [20]. Chi-square test was used to examine differences in *HLA-DRB1* and *HLA-DQB1* allele and haplotype frequencies (%) between MS and the healthy group. The odds ratio (OR) and 95% Confidence Interval (CI) were used to express the relative risk and *p* values under 0.05 were used to determine the significance of the results. In addition, following Chi-square analysis *P* value with less than 0.05, was corrected according to adjusted residual value. Bonferroni multiple comparison test was performed to correct the calculated *P* values. This was done in order to decrease Type I error [21]. Corrected *P* Values (*Pc*) below 0.013 was determined significant [22].

## 3. Results

### 3.1. Clinical characteristics of the volunteers

The age for MS patients involved in the study ranged from 19 to 58 years (mean age = 36.85 ± 9.5). The gender ratio representing female to male patients (F/M) was 1.5. The mean age of MS onset was 33.13 years. However, the average age of the control group was 26.17 years ± 6.02 and the gender ratio F/M was 0.66. The clinical parameters for the healthy volunteers and MS patients in this study are presented in Table 1.

### 3.2. *HLA-DRB1 and HLA-DQB1* alleles within MS patients and healthy control group

A significant difference between MS patients and healthy individuals in the following *HLA-DRB1 class II* alleles were observed: *HLA-DRB1 \*03:01* (*Pc* = 0.01) and *HLA-DRB1 \*04:01* (*Pc* = 0.004) are linked positively to MS, with a frequency of 19.2% and 5.4%, respectively,

**Table 1. Clinical characteristics of the volunteers.**

| Characteristics | Control (N = 75) | MS (N = 65) |
|---|---|---|
| Age (years) (mean ± SD) | 26.170 ± 6.020 | 36.850 ± 9.500 |
| Male n (%) | 45 (60%) | 26 (40%) |
| Female n (%) | 30 (40%) | 39 (60%) |
| Age of disease onset (years) (mean ± SD) | – | 33.130 ± 9.700 |
| General weakness n (%) | – | 60 (92.300%) |
| Optic neuritis n (%) | – | 23 (35.400%) |
| Sensory impairment n (%) | – | 28 (43.100%) |
| Ataxia n (%) | – | 14 (21.500%) |
| Brain steam symptom n (%) | – | 20 (30.800%) |
| Vertigo n (%) | – | 9 (13.800%) |
| Bladder dysfunction n (%) | – | 6 (9.200%) |
| Seizure n (%) | – | 7 (10.800%) |

compared to the control group, 8.7% and 0%, respectively (Table 2). While *HLA-DRB1 *04:03* (*Pc* = 0.001) allele showed a negative association with MS (Table 2).

HLA-DQB1 alleles' frequency is shown in Table 3. Our results presented no association in MS group compared to the control group, but *HLA-DQB1 *06:03 (Pc = 0.002)* may be suggested as protective allele since it linked to the control group.

### 3.3. *HLA-DRB1 ~ HLA-DQB1* haplotype within MS patients and healthy control group

For haplotypes, no parents' samples were accessible. The HLA haplotype approximation was accomplished by HaploStats software based on haplotype frequencies. Our results showed no statistical significance in *HLA-DRB1 ~ HLA-DQB1* haplotype in MS patients (Table 4). The following *HLA-DRB1 ~ HLA-DQB1* haplotypes showed statistical significance in control group in comparison to MS patients: *HLA-DRB1 *04:03 ~ HLA-DQB1 *03:02* (*Pc* = 0.004) and *HLA-DRB1 *13:01 ~ HLA-DQB1 *06:03* (*Pc* = 0.002) (Table 4).

### 3.4. Association between *HLA-DRB1 and HLA-DQB1* alleles and *HLA-DRB1 ~ HLA-DQB1* haplotype within MS patients (intra group)

The *HLA-DRB1 *03:01, HLA-DRB1 *11:01, and HLA-DRB1 *15:01* alleles were the most frequently observed in MS patients who have optic neuritis. However, a statistically significant association was observed among MS patients with optic neuritis in the *HLA-DRB1 *15:01* (*Pc* < 0.001) (S1 Table). Furthermore, *HLA-DQB1 *06:01* (*Pc* = 0.012) allele showed positive association with MS with optic neuritis, with a frequency of 10.9% in MS patients with optic neuritis and 2.4% in MS patients without optic neuritis (S1 Table). Additionally, our data showed no association in *HLA-DRB1 ~ HLA-DQB1* haplotype among MS group with optic

**Table 2.  *HLA-DRB1* alleles frequency of control group and MS patients among Jordanian population.**

| HLA-DRB1* | Control 2N = 150 | Allele Frequency (%) | MS 2N = 130 | Allele Frequency (%) | P | Pc | OR | 95% CI |
|---|---|---|---|---|---|---|---|---|
| 01:01 | 9 | 6. 000 | 2 | 1.500 | 0.055 | – | 0.245 | 0.052–1.154 |
| 01:03 | 2 | 1.300 | 0 | 0. 000 | 0.341 | – | 0.227 | 0.011–4.784 |
| **03:01** | **13** | **8.700** | **25** | **19.200** | **0.010** | **0.010** | **2.509** | **1.225–5.138** |
| **04:01** | **0** | **0. 000** | **7** | **5.400** | **0.047** | **0.004** | **18.279** | **1.034–323.245** |
| 04:02 | 3 | 2. 000 | 0 | 0. 000 | 0.229 | – | 0.1620 | 0.008–3.156 |
| **04:03** | **11** | **7.3**00 | **0** | **0. 000** | **0.034** | **0.001** | **0.047** | **0.003-0.797** |
| 07:01 | 26 | 17.300 | 11 | 8.500 | **0.029** | 0.441 | 0.441 | 0.209–0.932 |
| 08:01 | 3 | 2. 000 | 3 | 2.300 | 0.859 | – | 1.157 | 0.230–5.836 |
| 09:01 | 0 | 0. 000 | 1 | 0.800 | 0.446 | – | 3.487 | 0.141–86.329 |
| 10:01 | 8 | 5.300 | 2 | 1.500 | 0.088 | – | 0.277 | 0.058–1.330 |
| 11:01 | 22 | 14.700 | 28 | 21.500 | 0.134 | – | 1.597 | 0.863–2.957 |
| 11:02 | 8 | 5.300 | 10 | 7.700 | 0.422 | – | 1.479 | 0.566–3.867 |
| 12:01 | 1 | 0.700 | 5 | 3.800 | 0.067 | – | 5.960 | 0.687–51.688 |
| 13:01 | 13 | 8.700 | 5 | 3.800 | 0.069 | – | 0.389 | 0.136–1.110 |
| 13:02 | 0 | 0. 000 | 1 | 0.800 | 0.446 | – | 3.487 | 0.141–86.329 |
| 13:03 | 7 | 4.700 | 3 | 2.300 | 0.289 | – | 0.483 | 0.122–1.906 |
| 13:05 | 0 | 0. 000 | 1 | 0.800 | 0.446 | – | 3.487 | 0.141–86.329 |
| 14:01 | 8 | 5.300 | 2 | 1.500 | 0.880 | – | 0.277 | 0.058–1.330 |
| 14:04 | 1 | 0.700 | 0 | 0. 000 | 0.557 | – | 0.382 | 0.015–9.455 |
| 15:01 | 13 | 8.700 | 23 | 17.700 | **0.014** | 0.024 | 2.472 | 1.177–5.193 |
| 15:02 | 1 | 0.700 | 0 | 0. 000 | 0.577 | – | 0.382 | 0.015–3.523 |
| 16:01 | 1 | 0.700 | 0 | 0. 000 | 0.577 | – | 0.382 | 0.015–3.523 |
| 16:02 | 0 | 0. 000 | 1 | 0.800 | 0.446 | – | 3.487 | 0.141–86.329 |

N: number of volunteers, *Pc*: Corrected *P* value ≤ 0.013 OR: odds ratio, CI: Confidence Interval

**Table 3.  *HLA-DQB1* alleles frequency of control group and MS patients among Jordanian population.**

| HLA-DQB1* | Control 2N = 150 | Allele Frequency (%) | MS 2N = 130 | Allele Frequency (%) | P | Pc | OR | 95% CI |
|---|---|---|---|---|---|---|---|---|
| 02:01 | 33 | 22. 000 | 36 | 27.700 | 0.270 | – | 1.358 | 0.787–2.341 |
| 02:02 | 2 | 1.300 | 0 | 0. 000 | 0.341 | – | 0.228 | 0.011–4.784 |
| 03:01 | 40 | 26.700 | 33 | 25.400 | 0.807 | – | 0.936 | 0.548–1.599 |
| 03:02 | 17 | 11.300 | 11 | 8.500 | 0.424 | – | 0.723 | 0.326–1.606 |
| 03:03 | 3 | 2. 000 | 3 | 2.300 | 0.859 | – | 1.157 | 0.230–5.836 |
| 04:01 | 0 | 0. 000 | 3 | 2.300 | 0.164 | – | 8.263 | 0.423–161.480 |
| 04:02 | 2 | 1.300 | 0 | 0. 000 | 0.341 | – | 0.228 | 0.011–4.784 |
| 05:01 | 24 | 16. 000 | 13 | 10. 000 | 0.139 | – | 0.583 | 0.284–1.199 |
| 05:02 | 1 | 0.700 | 1 | 0.800 | 0.919 | – | 1.155 | 0.072–18.652 |
| 05:03 | 1 | 0.700 | 0 | 0. 000 | 0.557 | – | 0.382 | 0.0154–9.455 |
| 06:01 | 2 | 1.300 | 7 | 5.400 | 0.055 | – | 4.211 | 0.859–20.642 |
| 06:02 | 15 | 10. 000 | 23 | 17.700 | 0.061 | – | 1.935 | 0.962–3.889 |
| **06:03** | **10** | **6.700** | **0** | **0.000** | **0.041** | **0.002** | **0.051** | **0.003-0.884** |

N: number of volunteers, *Pc*: Corrected *P* value ≤ 0.013 OR: odds ratio, CI: Confidence Interval

**Table 4. The Frequency of *HLA-DRB1 ~ HLA-DQB1* haplotypes of control group and MS patients among Jordanian population.**

| *HLA- DRB1\* ~ HLA-DQB1\** | Control | | MS | | *P* | *Pc* | OR | 95% CI |
|---|---|---|---|---|---|---|---|---|
| | 2N = 150 | Haplotype Frequency (%) | 2N = 130 | Haplotype Frequency (%) | | | | |
| 01:01 ~ 03:01 | 2 | 1.300 | 0 | 0.000 | 0.075 | – | 0.753 | 0.953–1.543 |
| 01:01 ~ 05:01 | 7 | 4.700 | 2 | 1.500 | 0.075 | – | 0.073 | 0.004–1.296 |
| 01:03 ~ 03:01 | 2 | 1.300 | 0 | 0.000 | 0.341 | – | 0.228 | 0.011–4.784 |
| 03:01 ~ 02:01 | 11 | 7.300 | 19 | 14.600 | 0.049 | 0.050 | 2.163 | 0.988–4.734 |
| 03:01 ~ 03:01 | 0 | 0.000 | 2 | 1.500 | 0.255 | – | 5.856 | 0.279–123.100 |
| 03:01 ~ 03:02 | 0 | 0.000 | 1 | 0.800 | 0.446 | – | 3.487 | 0.141–86.329 |
| 03:01 ~ 03:03 | 1 | 0.700 | 0 | 0.000 | 0.557 | – | 0.382 | 0.015–9.455 |
| 03:01 ~ 05:01 | 1 | 0.700 | 0 | 0.000 | 0.557 | – | 0.382 | 0.015–9.455 |
| 03:01 ~ 06:02 | 0 | 0.000 | 2 | 1.500 | 0.255 | – | 5.856 | 0.279–123.100 |
| 04:01 ~ 03:01 | 0 | 0.000 | 3 | 2.300 | 0.164 | – | 8.263 | 0.423–161.479 |
| 04:01 ~ 03:02 | 0 | 0.000 | 3 | 2.300 | 0.164 | – | 8.263 | 0.423–161.479 |
| 04:01 ~ 06:01 | 0 | 0.000 | 1 | 0.800 | 0.446 | – | 3.487 | 0.141–86.329 |
| 04:02 ~ 03:02 | 3 | 2.000 | 0 | 0.000 | 0.229 | – | 0.162 | 0.083–3.155 |
| **04:03 ~ 03:02** | **9** | **6.000** | **0** | **0.000** | **0.049** | **0.004** | **0.057** | **0.003–0.990** |
| 04:03 ~ 04:02 | 2 | 1.300 | 0 | 0.000 | 0.341 | – | 0.228 | 0.011–4.784 |
| 07:01 ~ 02:01 | 24 | 16.000 | 9 | 6.900 | 0.027 | 0.019 | 0.411 | 0.183-0.923 |
| 07:01 ~ 03:01 | 0 | 0.000 | 1 | 0.800 | 0.446 | – | 3.487 | 0.141–86.329 |
| 07:01 ~ 03:03 | 2 | 1.300 | 1 | 0.800 | 0.647 | – | 0.574 | 0.051–6.400 |
| 08:01 ~ 03:01 | 3 | 2.000 | 1 | 0.800 | 0.387 | – | 0.380 | 0.039–3.697 |
| 08:01 ~ 03:02 | 0 | 0.000 | 1 | 0.800 | 0.446 | – | 3.487 | 0.141–86.329 |
| 08:01 ~ 04:01 | 0 | 0.000 | 1 | 0.800 | 0.446 | – | 3.487 | 0.141–86.329 |
| 09:01 ~ 03:03 | 0 | 0.000 | 1 | 0.800 | 0.446 | – | 3.487 | 0.141–86.329 |
| 10:01 ~ 03:02 | 2 | 1.300 | 0 | 0.000 | 0.341 | – | 0.228 | 0.011–4.784 |
| 10:01 ~ 05:01 | 6 | 4.000 | 2 | 1.500 | 0.218 | – | 0.375 | 0.074–1.891 |
| 11:01 ~ 02:01 | 0 | 0.000 | 4 | 3.100 | 0.113 | – | 10.708 | 0.571–200.796 |
| 11:01 ~ 03:01 | 20 | 13.300 | 18 | 13.800 | 0.901 | – | 1.045 | 0.527–2.073 |
| 11:01 ~ 03:02 | 1 | 0.700 | 2 | 1.500 | 0.480 | – | 2.328 | 0.209–25.974 |
| 11:01 ~ 05:01 | 0 | 0.000 | 4 | 3.100 | 0.113 | – | 10.708 | 0.571–200.796 |
| 11:01 ~ 05:02 | 0 | 0.000 | 1 | 0.800 | 0.466 | – | 3.487 | 0.141–86.329 |
| 11:01 ~ 06:02 | 1 | 0.700 | 1 | 0.800 | 0.919 | – | 1.155 | 0.072–18.653 |
| 11:02 ~ 02:01 | 0 | 0.000 | 1 | 0.800 | 0.466 | – | 3.487 | 0.141–86.329 |
| 11:02 ~ 03:01 | 7 | 4.700 | 3 | 2.300 | 0.289 | – | 0.483 | 0.122–1.906 |
| 11:02 ~ 06:02 | 1 | 0.700 | 5 | 3.800 | 0.067 | – | 5.960 | 0.687–51.688 |
| 12:01 ~ 02:01 | 0 | 0.000 | 2 | 1.500 | 0.255 | – | 5.856 | 0.279–123.100 |
| 12:01 ~ 03:01 | 1 | 0.700 | 2 | 1.500 | 0.480 | – | 2.328 | 0.209–25.974 |
| 12:01 ~ 06:02 | 0 | 0.000 | 1 | 0.800 | 0.466 | – | 3.487 | 0.141–86.329 |
| 13:01 ~ 04:01 | 0 | 0.000 | 1 | 0.800 | 0.466 | – | 3.487 | 0.141–86.329 |
| 13:01 ~ 05:01 | 3 | 2.000 | 1 | 0.800 | 0.387 | – | 0.380 | 0.039–3.697 |
| 13:01 ~ 06:02 | 0 | 0.000 | 3 | 2.300 | 0.164 | – | 8.263 | 0.423–161.479 |
| **13:01 ~ 06:03** | **10** | **6.700** | **0** | **0.000** | **0.041** | **0.002** | **0.051** | **0.003-0.884** |
| 13:02 ~ 05:01 | 0 | 0.000 | 1 | 0.800 | 0.446 | – | 3.487 | 0.141–86.329 |
| 13:03 ~ 03:01 | 5 | 3.300 | 3 | 2.300 | 0.607 | – | 0.685 | 0.161–2.923 |
| 13:03 ~ 03:02 | 2 | 1.300 | 0 | 0.000 | 0.341 | – | 0.228 | 0.011–4.784 |
| 13:05 ~ 03:01 | 0 | 0.000 | 1 | 0.800 | 0.446 | – | 3.487 | 0.141–86.329 |
| 14:01 ~ 05:01 | 0 | 0.000 | 1 | 0.800 | 0.466 | – | 3.487 | 0.141–86.329 |

*(Continued)*

**Table 4.** (Continued)

| | Control | | MS | | | | | |
|---|---|---|---|---|---|---|---|---|
| 14:01 ~ 06:02 | 0 | 0.000 | 1 | 0.800 | 0.466 | – | 3.487 | 0.141–86.329 |
| 15:01 ~ 03:02 | 0 | 0.000 | 4 | 3.100 | 0.113 | – | 10.708 | 0.571–200.796 |
| 15:01 ~ 05:01 | 0 | 0.000 | 2 | 2.500 | 0.255 | – | 5.856 | 0.279–123.100 |
| 15:01 ~ 06:01 | 0 | 0.000 | 5 | 3.800 | 0.082 | – | 0.685 | 0.161–2.924 |
| 15:01 ~ 06:02 | 0 | 0.000 | 12 | 9.200 | **0.017** | 0.180 | 31.751 | 1.861–541.807 |
| 16:02 ~ 03:01 | 0 | 0.000 | 1 | 0.800 | 0.466 | – | 3.487 | 0.141–86.329 |

N: number of volunteers, $Pc$: Corrected $P$ value ≤ 0.013 OR: odds ratio, CI: Confidence Interval

neuritis (S2 Table). While the *HLA-DRB1 *03:01, HLA-DRB1 *11:01, and HLA-DRB1 *15:01* alleles were the most frequently observed in MS patients with sensory impairment, but they do not reach a statistically significant association (S3 Table). Moreover, *HLA-DQB1 *05:01* and *HLA-DQB1 *06:02* alleles also demonstrated a positive correlation with MS with sensory impairment with $Pc$ values of $Pc < 0.001$ and $Pc = 0.006$, respectively (S3 Table). Additionally, *HLA-DRB1 *07:01* allele indicated a positive correlation in MS with brainstem symptoms with $Pc$ value < 0.001 (S4 Table).

## 4. Discussion

MS is a chronic inflammatory disease that impairs the CNS function of young people. This work is considered the first in terms of studying the association of HLA class II genes among Jordanian MS patients compared to healthy individuals. In addition, we studied the association of HLA class II genes in MS patients with different clinical features, such as optic neuritis, sensory impairment, and brainstem symptoms. Our findings showed a significant association between MS patients and *HLA-DRB1 *03:01* and *HLA-DRB1 *04:01* alleles in comparison to the control healthy group, which indicate that these alleles are associated with MS disease among Jordanians. On the contrary, the suggested protective *HLA-DRB1* allele is *HLA-DRB1 *04:03* as well as *HLA-DQB1 *06:03* allele.

Regarding the association of *HLA-DRB1 *03:01* with Jordanian MS patients, this finding aligns with several previous studies that has been conducted in Sardinian, Australian, and African-American populations [23–25]. Moreover, our study has demonstrated that *HLA-DRB1 *0401* allele influence MS susceptibility in Jordanian patients, which is similar to other studies being performed in Australian and Caucasians populations [24,26]. In addition, our results revealed strong association between *HLA-DRB1 *15:01* allele and MS with optic neuritis. This result is comparable with other studies that have been conducted consistently in different populations from Europe, Africa, and Latin America. They have shown that MS is associated with *HLA-DRB1 *15:01* allele of HLA class II gene, as well as *HLA-DRB1 *15* displayed a stronger association among Americans [27–29].

Furthermore, several studies have been conducted in the MENA countries, such as the Arabian Gulf, Iranian, Tunisian, and Saudi MS patients, and showed that *HLA-DRB1 *15:01* is high risk with MS disease [30–33]. In contrast, a study showed no association between *HLA-DRB1 *15:01* and MS among Bahraini patients [34]. Also, our results presented that *HLA-DRB1 *04:03* has protective effects for MS disease among Jordanians. On the other hand, previous work from Sweden and Italy have demonstrated the protective effects of *HLA-DRB1 *07:01* [6,35]. Additionally, regarding *HLA-DQB1* alleles, our results indicated that *HLA-DQB1 *06:03* allele has protective effect to MS disease since this allele was completely absent among MS patients. In addition, regarding HLA haplotype level, according to earlier

work in Caucasians that has shown that HLA class II haplotype with the strongest correlation with MS is *HLA-DRB1 *15:01 ~ HLA-DQB1 *06:02* [31]. Our results revealed no association on the haplotype level with MS disease.

Moreover, an intra-group analysis in our study was conducted to show which HLA class II alleles contribute to MS clinical features susceptibility or have protective effect against these features. This was achieved by comparing the expression of HLA subtypes alleles between MS patients based on their clinical features. The analysis showed that a significant link between optic neuritis in MS patients and *HLA-DRB1 *15:01* and *HLA-DQB1 *06:01* alleles compared to MS group without optic neuritis. Similarly, the same result was obtained in Japanese MS patients [36]. However, Deschamps *et al.* reported that *HLA-DRB1 *15* did not significantly affect the risk of optic neuritis in French population, compared to the control group, although it was linked to a higher susceptibility to MS [37]. While in other MS populations in the Netherlands and India demonstrated that *HLA-DRB1 *03:01* allele is linked with optic neuritis [38–40]. Several HLA class II alleles has also been shown to be associated with optic neuritis, such as *HLA-DRB1 *16:02* in China and *HLA-DRB1 *04:05* in South Brazil [26,41]. Also, our study revealed a significant association between the sensory impairment and *HLA-DQB1 *05:01* and *HLA-DQB1 *06:02* alleles but there is no linkage with *HLA-DRB1* alleles. While *HLA-DRB1 *01:01* allele decreased the spinal cord involvement and sensory impairment manifestation in Japanese MS patients, a reduction of brainstem symptoms was linked to *HLA-DRB1 *09:01* and *HLA-DRB1 *13:02* [36]. On the other hand, our results suggested that *HLA-DRB1 *07:01* is positively associated with brainstem symptoms in Jordanian MS patients. Genetic ethnic diversity along with differences in populations and geographic areas may be attributed to the variations in the results. All in all, we admit that the current study has some limitations, such as small sample size and the variation in gender distribution among the studied groups. Nevertheless, to the best of our knowledge, this is the first study in Jordan, to highlight the influence of HLA class II genes in Jordanian MS patients. However, further studies with larger sample size are needed. In addition, to best of our knowledge, there are no studies that have reported the difference in association between HLA genes and gender in MS patients, hence, future studies to include such investigations between males and females would be useful. So far, the results of our study can be used as preliminary study until replicated in a larger sample size in the future work.

## 5. Conclusions

In conclusion, *HLA-DRB1 *03:01, HLA-DRB1 *04:01* alleles are reported as risk factors for Jordanians MS patients, according to our analysis of the HLA class II immunogenetic profile of MS patients in Jordan. These alleles could therefore be used as potential predictive indicators for early MS risk screening. Furthermore, these alleles have positive association with different autoimmune diseases among Jordanians, such as systemic lupus erythematous and juvenile diabetes as was reported in our previous studies [42,43]. Moreover, there are different HLA class II genes among Jordanians MS patients that are linked with several MS manifestations, such as: the *HLA-DRB1 *15:01* and *HLA-DQB1 *06:01* alleles increase the likelihood of optic neuritis, *HLA-DRB1 *07:01* is linked to brainstem symptoms, and sensory impairment was linked with *HLA-DQB1 *05:01* and *HLA-DQB1 *06:02* alleles. Knowing how these genotypes work may also help in the development of more potent therapy regimens for MS and its related problems, such as optic neuritis, sensory impairment, and brainstem symptoms. Understanding the etiology and progression of MS is important for the prevention, diagnosis, and clinical management of aforementioned disorders associated with MS. It is necessary to study additional genetic markers that are thought to be candidate risk genes for Jordanian MS patients in order to further understand the disease's pathogenesis.

## Supporting Information

**S1 Table. Frequency of *HLA-DRB1* and *HLA-DQB1* allele among MS patients with optic neuritis and without optic neuritis.** N: number of volunteers, *Pc*: Corrected *P* value ≤ 0.013 OR: odds ratio, CI: Confidence Interval.
(DOCX)

**S2 Table. The frequency of *HLA-DRB1* ~ *HLA-DQB1* haplotype among MS patients with and without optic neuritis in Jordanian population.** N: number of volunteers, *Pc*: Corrected *P* value ≤ 0.013 OR: odds ratio, CI: Confidence Interval.
(DOCX)

**S3 Table. The frequency of *HLA-DRB1* and *HLA-DQB1* alleles among MS patients with sensory impairment and without sensory impairment.** N: number of volunteers, *Pc*: Corrected *P* value ≤ 0.013 OR: odds ratio, CI: Confidence Interval.
(DOCX)

**S4 Table. Frequency of *HLA-DRB1* alleles among MS patients with brainstem symptoms and without brainstem symptoms.** N: number of volunteers, *Pc*: Corrected *P* value ≤ 0.013 OR: odds ratio, CI: Confidence Interval.
(DOCX)

## Acknowledgments

The authors would like to thank all the volunteers who took apart in this study, as well as all the team in neuro-medicine clinic at Al-Bashir Hospital, Amman for their help in collecting the samples.

## Author contributions

**Conceptualization:** Sawsan I. Khdair.

**Formal analysis:** lubna Al-Khareishaa, Alaa M. Hammad, Alaa Khudair.

**Funding acquisition:** Sawsan I. Khdair, Alaa M. Hammad.

**Investigation:** Sawsan I. Khdair, lubna Al-Khareishaa, Alaa Khudair.

**Methodology:** Sawsan I. Khdair, lubna Al-Khareishaa.

**Writing – original draft:** Sawsan I. Khdair, Osama H. Abusara.

**Writing – review & editing:** Sawsan I. Khdair, lubna Al-Khareishaa, Osama H. Abusara, Alaa M. Hammad.

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
