## [Decision Letter · Decision Letter 0]

14 Jan 2025

PONE-D-24-50217HLA-Class II Genes Association with Multiple Sclerosis: An Immunogenetic Prediction Among Multiple Sclerosis Jordanian PatientsPLOS ONE

Dear Dr. Khdair,

Thank you for submitting your manuscript to PLOS ONE. After careful consideration, we feel that it has merit but does not fully meet PLOS ONE’s publication criteria as it currently stands. Therefore, we invite you to submit a revised version of the manuscript that addresses the points raised during the review process.

We look forward to receiving your revised manuscript.

Kind regards,

Sreeram V. Ramagopalan

Academic Editor

PLOS ONE

“the project was fully funded from Al-Zaytoonah University of Jordan for  (Grants numbers: 31/11/2020-2021 and 58/17/2022-2023).”

“The authors would like to thank all the volunteers who took apart in this study, as well as all the team in neuro-medicine clinic at Al-Bashir Hospital, Amman for their help in collecting the samples. The authors would also thank Al-Zaytoonah University of Jordan for financially supporting the project (Grants numbers: 31/11/2020-2021 and 58/17/2022-2023).”

“the project was fully funded from Al-Zaytoonah University of Jordan for  (Grants numbers: 31/11/2020-2021 and 58/17/2022-2023).”

Reviewers' comments:

Reviewer's Responses to Questions

**Comments to the Author**

1. Is the manuscript technically sound, and do the data support the conclusions?

Reviewer #1: Yes

Reviewer #2: Partly

2. Has the statistical analysis been performed appropriately and rigorously? 

Reviewer #1: Yes

Reviewer #2: No

3. Have the authors made all data underlying the findings in their manuscript fully available?

Reviewer #1: Yes

Reviewer #2: No

4. Is the manuscript presented in an intelligible fashion and written in standard English?

Reviewer #1: Yes

Reviewer #2: Yes

5. Review Comments to the Author

Reviewer #1: Summary and general comments:

In the manuscript “HLA-Class II Genes Association with Multiple Sclerosis: An Immunogenetic Prediction Among Multiple Sclerosis Jordanian Patients” the authors took a correlation approach to assess the association between HLA class II genes and Multiple Sclerosis (MS). To achieve such aim, HLA class II genotyping by PCR SSP was performed in 65 MS patients comparatively to 75 healthy controls.

The main results showed:

1. Significant association of HLA-DRB1*03:01 and DRB1*0401 alleles with MS susceptibility

2. Significant association of HLA-DRB1*15:01 and DQB1*06:01 alleles with optic neuritis

Overall, the paper is well organized and I really enjoyed reading it. I believe it fits for publication in its current form.

Reviewer #2: Khdair and co-authors examined the distribution of HLA-DRB1 and -DQB1 alleles in a cohort of Jordanian patients with multiple sclerosis (MS), comparing their findings to a previously studied, similarly HLA-typed group of Jordanian controls (references 41 and 42). The manuscript also reviewed recent and comparable studies on MS and HLA alleles.

A key concern was the application of the Bonferroni correction to calculate Pc values. The authors used a fixed p-value of 0.013 to determine significant differences; however, after applying the Bonferroni correction, no alleles showed a significantly different frequency compared to the controls. The relatively small number of patients and controls is a limitation of the study. Additionally, the authors could enhance their analysis by incorporating data from broader population sources, such as NEMA, using resources like AlleleFrequencies.net.

Minor suggestion: Ensure uniformity in the decimal values presented in the tables.

6. PLOS authors have the option to publish the peer review history of their article (what does this mean? ). If published, this will include your full peer review and any attached files.

**Do you want your identity to be public for this peer review?** For information about this choice, including consent withdrawal, please see our Privacy Policy .

Reviewer #1: **Yes: ** Tarak Dhaouadi

Reviewer #2: No

---

## [Author Response · Author response to Decision Letter 0]

17 Jan 2025

Attached in the "Response to Reviewers" file and added here as well:

Editor-In-Chief, PLOS ONE

Reference No.: PONE-D-24-50217

Title: HLA-Class II Genes Association with Multiple Sclerosis: An Immunogenetic Prediction Among Multiple Sclerosis Jordanian Patients

Dear Editor-In-Chief:

We appreciate the helpful suggestions and constructive comments provided by the reviewers. Revision was requested. Please find a revised version of our manuscript entitled “HLA-Class II Genes Association with Multiple Sclerosis: An Immunogenetic Prediction Among Multiple Sclerosis Jordanian Patients”. We have revised the manuscript in response to the reviewers’ comments in the first round. Changes in the submitted manuscript were highlighted in red. Changes were also made accordingly in the Supporting Information file wherever required (not highlighted).

As for the journal requirements, please note the following:

1. “The Project was fully funded by Al-Zaytoonah University of Jordan grants (Grants numbers: 31/11/2020-2021 and 58/17/2022-2023).”

2. “The funder had no role in study design, data collection and analysis, decision to publish, or preparation of the manuscript.”

3. Data availability statement: “All data generated from the study are presented within the manuscript and the supporting information.”

Please find below our responses to the reviewers’ comments.

Reviewer #1:

Summary and general comments:

In the manuscript “HLA-Class II Genes Association with Multiple Sclerosis: An Immunogenetic Prediction Among Multiple Sclerosis Jordanian Patients” the authors took a correlation approach to assess the association between HLA class II genes and Multiple Sclerosis (MS). To achieve such aim, HLA class II genotyping by PCR SSP was performed in 65 MS patients comparatively to 75 healthy controls.

The main results showed:

1. Significant association of HLA-DRB1*03:01 and DRB1*0401 alleles with MS susceptibility

2. Significant association of HLA-DRB1*15:01 and DQB1*06:01 alleles with optic neuritis

Overall, the paper is well organized and I really enjoyed reading it. I believe it fits for publication in its current form.

We thank the reviewer for the comments.

Reviewer #2:

Khdair and co-authors examined the distribution of HLA-DRB1 and -DQB1 alleles in a cohort of Jordanian patients with multiple sclerosis (MS), comparing their findings to a previously studied, similarly HLA-typed group of Jordanian controls (references 41 and 42).

The manuscript also reviewed recent and comparable studies on MS and HLA alleles.

We thank the reviewer for the comment.

A key concern was the application of the Bonferroni correction to calculate Pc values. The authors used a fixed p-value of 0.013 to determine significant differences; however, after applying the Bonferroni correction, no alleles showed a significantly different frequency compared to the controls.

We thank the reviewer for the comment. Since one limitation of the study, having a small sample size (N = 65 in MS patients and N = 75 in control), p-value with correction was needed to decrease Type I error. We added “This was done in order to decrease Type I error [21].” in statistical analysis lines 132-133. Although following the correction some alleles showed non-significant different in frequency, we reported both p values before and after correction. Please note that the number of references were also updated accordingly within the manuscript.

The relatively small number of patients and controls is a limitation of the study. Additionally, the authors could enhance their analysis by incorporating data from broader population sources, such as NEMA, using resources like AlleleFrequencies.net.

We thank the reviewer for the comment, we mention this limitation following the discussion. According to AlleleFrequencies.net there is no HLA study concerning MS was conducted in Jordan. This was mentioned in the discussion section as this is the first study on HLA association with MS to be conducted in Jordan.

Minor suggestion: Ensure uniformity in the decimal values presented in the tables.

We thank the reviewer for the comment and we updated the tables.

---

## [Editor Report · Decision Letter 1]

22 Jan 2025

HLA-Class II Genes Association with Multiple Sclerosis: An Immunogenetic Prediction Among Multiple Sclerosis Jordanian Patients

PONE-D-24-50217R1

Dear Dr. Khdair,

We’re pleased to inform you that your manuscript has been judged scientifically suitable for publication and will be formally accepted for publication once it meets all outstanding technical requirements.

Kind regards,

Sreeram V. Ramagopalan

Academic Editor

PLOS ONE
---

## [Editor Report · Acceptance letter]

PONE-D-24-50217R1

PLOS ONE

Dear Dr. Khdair,

I'm pleased to inform you that your manuscript has been deemed suitable for publication in PLOS ONE. Congratulations! Your manuscript is now being handed over to our production team.

Kind regards,

on behalf of

Dr. Sreeram V. Ramagopalan

Academic Editor

PLOS ONE